# Selected Physico-Chemical, Nutritional, Antioxidant and Sensory Properties of Wheat Bread Supplemented with Apple Pomace Powder as a By-Product from Juice Production

**DOI:** 10.3390/plants11091256

**Published:** 2022-05-06

**Authors:** Veronika Valková, Hana Ďúranová, Michaela Havrlentová, Eva Ivanišová, Ján Mezey, Zuzana Tóthová, Lucia Gabríny, Miroslava Kačániová

**Affiliations:** 1AgroBioTech Research Centre, Slovak University of Agriculture, Trieda Andreja Hlinku 2, 94976 Nitra, Slovakia; veronika.valkova@uniag.sk (V.V.); hana.duranova@uniag.sk (H.Ď.); zuzana.tothova@uniag.sk (Z.T.); lucia.gabriny@uniag.sk (L.G.); 2Institute of Horticulture, Slovak University of Agriculture, Trieda Andreja Hlinku 2, 94976 Nitra, Slovakia; jan.mezey@uniag.sk; 3National Agricultural and Food Centre, Research Institute of Plant Production, Bratislavska cesta 122, 92168 Piešťany, Slovakia; michaela.havrlentova@ucm.sk; 4Department of Biotechnologies, Faculty of Natural Sciences, University of Ss. Cyril and Methodius, Nám. J. Herdu 2, 91701 Trnava, Slovakia; 5Institute of Food Sciences, Slovak University of Agriculture, Trieda Andreja Hlinku 2, 94976 Nitra, Slovakia; eva.ivanisova@uniag.sk; 6Department of Bioenergy, Food Technology and Microbiology, Institute of Food Technology and Nutrition, University of Rzeszow, 4 Zelwerowicza Str., 35-601 Rzeszow, Poland

**Keywords:** bakery product, apple pomace, nutritional profile, total polyphenol content, antioxidant activity, sensory properties

## Abstract

The present article aimed to study the effects of four selected concentrations (1%, 2%, 5%, and 10%) of apple pomace powder (APP), obtained from juice production, on the nutritional value and selected physico-chemical, antioxidant, and sensory properties of wheat bread. We have found that the ash and total carbohydrate contents, total polyphenols content, and antioxidant activity of the supplemented bread loaves were markedly higher (*p* < 0.05) as compared to the control ones. On the other hand, values for protein and fat contents and loaf volume in APP-containing bread samples were statistically lower (*p* < 0.05). Finally, sensory evaluation revealed no significant differences in all tested attributes between the investigated groups of bread samples. The current results suggest that 10% APP addition appears to be an attractive ingredient applied to bread formulation to obtain a bakery product with high nutritional value and required qualitative and sensory properties. In such a manner, apple pomace as by-products from apple juice processing can be efficiently utilized in an eco-friendly way by the food industry to decrease unnecessary waste and environmental pollution.

## 1. Introduction

In recent years, there has been a general trend towards reutilizing industrial by-products that would be otherwise thrown away as a waste [1]. Among them, a large amount of residue (representing around 25% of the original fruits) is generated by apple juice processing [2]. This apple industry by-product, known as apple pomace (AP), consists basically of 94.1% peels, 4.1% seeds, and 1.8% pulp [3]. Overall, several million metric tons of AP are estimated to be produced worldwide every year. Hence, the use of these waste products in other applications could lead to the elimination of environmental burdens, and so to reduction of public health hazards [4]. In addition, reusing of agro-industrial by-products is vital of importance for economic development and sustainability [5].

In terms of nutritional description, fresh AP contains considerable amounts of phytochemicals, such as carbohydrates, simple sugars, small amounts of proteins, vitamins, and minerals [2]. Furthermore, these by-products are also a rich source of functional biological active substances, especially polyphenols and natural antioxidants [6]. The nutritional profile of apple pomace powder (APP) obtained by drying reveals a high content of carbohydrate and fermentable sugar (up to 50%). Considering the proximate composition, its value in APP varied in the following manner: ash (0.50–6.10%) < fat (1.20–3.90%) < protein (2.94–5.67%) < pectin (3.50–14.32%) < moisture (3.90–10.80%) < fibre (4.70–51.10%) < soluble fibre (14.60%) < insoluble fibre (36.50%) < total carbohydrate (48.0–62.0%) [2].

Regarding the high-potential source of bioactive compounds, APP is intensively incorporated in diverse bakery products including cakes, muffins, cookies, bread, biscuits, crackers, and extruded snacks [1,7]. In effect, the consumption of these products comprising many antioxidants could eliminate oxidative stress, and thus, prevent various chronic diseases [8]. Since wheat bread is considered a worldwide staple product consumed by most of the population, its enrichment with APP can help to enhance AP utilization in the food industry on a large scale [9]. However, it is necessary to keep in mind that application of fruit by-products in bread formulation generally decreases their qualitative characteristics [10]. The negative effects of APP on bread quality are commonly related to lower volume and color modification, which ultimately results in increased hardness and unfavorable external appearance of bread loaves evidenced by consumers. Moreover, the presence of polyphenols, as well as other biological active substances in APP, is generally associated with bitter, acrid, or astringent taste, as the substances conventionally protect the fruits against herbivores [11]. In this line, the sensory evaluation performed by Garvey et al. [12] revealed a reduced odor and sweet flavor of sponge cakes related to the addition of APP.

As mentioned above, there are many studies dealing with the use of AP in various bakery products (including bread) [13,14,15,16]. However, the composition and functional properties of APP, consequently highly affecting the product quality, is strongly dependent on many diverse factors such as variety [6], ripening degree [10], climatic growing conditions, agricultural practices, storage, and suffered infections of apples [17], as well as apple processing [18] and AP processing (drying) methods [2,19]. Taking into account all of these aspects, discrepancies in the data obtained from various experiments have been still occurring, and so, each of them can serve as a framework for further scientific activities.

In this sense, the aim of the present study was to evaluate the impact of different concentrations (0%, 1%, 2%, 5%, and 10%) of APP obtained from our experimental plots on the qualitative characteristics of wheat bread. Specifically, nutritional profile, volume and sensory properties of the enriched bread loaves were investigated to accomplish our research objectives. Based on the analyses, the most adequate concentration of APP used for bread supplementation to meet health-promoting nutrition with the preservation of acceptable physical and sensory properties of bread loaves will be determined and recommended. Most importantly, the study was concurrently carried out as a part of more comprehensive research linked to apple juice production. Thus, the main goal of our study was not only to improve nutritional quality of bread but also to efficiently utilize the waste product in order to eliminate its amount in the environment.

## 2. Results

### 2.1. Nutritional Analysis of the Apple Pomace Powder and Wheat Flour

As was expected, the APP obtained from mixed apple varieties (Table 1) contained a low level of moisture (6.55 ± 0.16%), ash (1.65 ± 0.08%), protein (2.18 ± 0.09%), fat (1.58 ± 0.01%), and a high amount of carbohydrates (88.04 ± 0.09%). As compared to the WF, the APP showed significantly (*p* < 0.05) lower values for moisture, crude protein, and fat. On the other hand, their contents of ash and total carbohydrates together with energetic value were considerably (*p* < 0.05) higher. Furthermore, the TPC and AA of the APP were found to be markedly (*p* < 0.05) higher in comparison with the WF.

### 2.2. Nutritional Analysis of the Bread Loaves

The results from nutritional analysis including proximate properties, TPC, and antioxidant capacity are shown in Table 2. From them, it is evident that the APP addition did not have any significant effect (*p* < 0.05) on the moisture of the dried bread samples. Next, the ash and total carbohydrates in all supplemented bread loaves were demonstrably (*p* < 0.05) higher as compared to the control bread. Moreover, the values gradually increased with the increasing APP concentrations, and the differences between individual experimental groups of samples were statistically significant (*p* < 0.05), except for ash in the bread supplemented with 1% and 2% APP. On the other hand, values for protein and fat contents in the supplemented bread samples were statistically (*p* < 0.05) lower in comparison with the wheat bread. In both parameters, significant differences (*p* < 0.05) were also observed between all experimental groups of bread, except for those enriched with 1% and 2% APP.

Regarding the energetic value of the analyzed samples, notable differences (*p* < 0.05) were recorded between all experimental bread loaves in comparison with the control, and as compared to each other, as well. Indeed, the highest value for energy was found in the control sample, and the lowest ones in the samples with 5% and 10% addition.

From the obtained data it can be noted that the supplementation of APP in wheat bread significantly (*p* < 0.05) increased the content of total polyphenols, and AA as compared to the control (Table 2). Moreover, marked differences (*p* < 0.05) were recorded between experimental samples with gradually increasing APP addition. In effect, the bread loaves enriched with 10% APP had a 4.27-fold and a 1.7-fold increase in TPC and AA, respectively.

### 2.3. Bread Volume Analysis

From the results shown in Table 3 it is evident that the APP additions (1%, 2%, 5%, and 10%) into the WF led to a significant decrease (*p* < 0.05) in the volume of the bread samples as compared to the control one. Moreover, our results showed that the volumes of the experimental bread loaves were remarkably (*p* < 0.05) different from each other, and that they decreased with increasing APP proportion.

### 2.4. Sensory Evaluation of the Bread Loaves

The attractiveness of the control and APP-supplemented breads was examined with respect to sensory properties (shape and overall appearance, crust surface and properties, overall appearance of crumb, aroma, taste, and overall acceptability) in order to determine the maximum acceptable concentration of the APP addition. Figure 1 demonstrates radar graphs of the sensory data in the breads with APP contents at five different levels: 0%, 1%, 2%, 5%, and 10%. Sensory evaluation showed no significant differences in all tested attributes between the investigated groups of bread samples. Specifically, all tested bread samples excelled with harmonic, pleasant characteristics. Moreover, crust surface and properties, aroma, and overall acceptability in bread samples with up to 5% APP addition were rated by the panellists, with even higher scores than the control sample.

## 3. Discussion

The chemical composition of powder obtained from AP has been investigated in numerous experimental studies. Generally, it is known that APP is a rich source of carbohydrates [20], and contains small amounts of proteins and minerals [2], which is in line with our findings. O’Shea et al. [21] recorded that APP consisted of low fat (2.27 ± 0.23%), protein (2.37 ± 0.12%), and ash (1.60 ± 0.02%), which was also confirmed by our analyses (1.58 ± 0.01%, 2.18 ± 0.09%, 1.65 ± 0.08%, respectively). Moreover, the value for carbohydrates (84.76 ± 0.56%) detected by the authors was very similar to ours (88.04 ± 0.09%). From them, the AP carbohydrates consisted mainly of insoluble sugars including cellulose (7.2–43.6% *w*/*w*), hemicellulose (4.26–24.40% *w*/*w*), and lignin (15.3–23.5% *w*/*w*), together with simple sugars including glucose (22.7%), fructose (23.6%), arabinose (14–23%), and galactose (6–15%) [22]. The high content of total carbohydrates in our APP is linked to a high amount of cell wall carbohydrates consisting of apple peels [2], which were used for the APP production in our study.

In general, it is also well-known that AP has a high content of polyphenols with antioxidant activities [23]. However, their amounts are strongly dependent not only on the apple cultivars [6] but also on methods used for drying of plant material to produce a powder [18,24]. Our finding suggests that despite drying, the APP displayed a high content of total polyphenols (9.53 ± 0.03 g kg^−1^ GAE) together with a strong AA (7.70 ± 0.17 g kg^−1^ TEAC). In agreement with our results, the high TPC (5.27 mg catechin/g dw; methods without using F-C reagent), as well as AA (45.34 μMTx/kg dw; TEAC2), though evaluated by other methods, were also recorded by Gumul et al. [25]. Using three different drying techniques (oven, sun, and freeze drying) to remove moisture content from fresh AP, Rana et al. [19] evaluated the TPC and AA of the dehydrated samples of these AP from which the phenolics were extracted using three different extraction solvents (50% methanol, 50% acetone, and 50% ethanol). Their findings revealed that the values for TPC (1.00–3.31 g kg^−1^ GAE) and AA (2.09–3.74 g kg^−1^ TEAC) were lower as compared to our data. The differences between our study and theirs may be related to the non-identical methodological procedures used. Additionally, the different apple cultivars employed in the experiments may also explain these discrepancies in measured data. Based on the results, we can predict that consumption of our APP and APP-enriched products could eliminate oxidative stress and prevent various chronic diseases as a result of the biological active properties.

To find the consensus that the non-traditional addition in wheat bread as a potential source of natural antioxidants and macronutrients providing pro-health value to humans can also provide adequate qualitative characteristics to the staple bakery product consumed worldwide, the bread loaves with APP addition (1%, 2%, 5%, and 10%) were consequently evaluated from this point of view.

Among the macronutrients, total carbohydrates created the largest proportion in our control sample (61.65%). This fact is attributed to the presence of WF in the bread formulation which consisted of 73.47 ± 0.1% carbohydrates. In the experimental bread loaves with APP addition, values for carbohydrates ranged between 62.83 ± 0.08% (1% APP) to 67.35 ± 0.21% (10% APP). These higher values are commonly linked to the high carbohydrate amounts in the APP itself (approximately 88.04 ± 0.09%). Higher content of proteins (11.78 ± 0.13%) and lower amounts of ash (0.61 ± 0.05%) in the WF than in the APP (2.18 ± 0.09%, 1.65 ± 0.08%, respectively; Table 1) were reflected by gradually lower protein content and higher ash in the experimental bread loaves, as expected. In accordance with our findings, Usman et al. [15] also found decreased protein content (5.34 ± 0.70%) and increased ash (0.62 ± 0.02%) in cookies with increasing levels (0%, 5%, 10%, 15%, and 20%) of APP supplementation as compared to the control sample (6.59 ± 0.47%, 0.45 ± 0.03, respectively).

Furthermore, our data showing significantly higher values for TPC and AA in the APP-containing bread samples than the control one was also demonstrated by Gumul et al. [25]. Similarly, Drożdż et al. [26], focusing on enrichment of extruded snacks with APP, also revealed their increased values for both evaluated parameters. Considering the existence of positive correlation between TPC and AA [27] we suppose that the increased antioxidant capacity in our bread samples supplemented with APP can be partially due to the high TPC in the powder; however, other compounds with antioxidant properties (such as vitamin C, E, beta-carotene), and minerals (that were not analyzed in our study) can also contribute to strong AA identified in our bread samples.

In summary, our results pointed to the possibility of bread loaves production with a rich nutritional profile, high amounts of phenolic substances, as well as strong antioxidant properties (improving human health) through addition of the APP obtained as an agro-industrial waste from apple juice extraction.

As mentioned above, the addition of APP in bakery goods is beneficial for improving nutritional values. However, there are also some challenges associated with APP supplementation, such as deterioration of physical properties including volume. Determination of the loaf volume is an important aspect reflecting the bread quality that provides a qualitative measurement of baking performance. In our study, we observed a decrease in this parameter with an increase in the APP levels. When it comes to the adverse impact of by-product powder on the bread volume, the same tendency was documented by Šporin et al. [28]. The authors noticed a negative effect of grape pomace powder on this parameter which was substantially decreased with increasing concentration of the powder addition (0%, 6%, 10%, and 15%). Similarly, Sudha et al. [29] observed a gradual reduction in cake volume with APP (10%, 20% and 30%) supplementation. In this case, the researchers detected up to a 27.06% decrease in the volume of cake enriched with 30% APP addition as compared to the control. Moreover, Torbica et al. [30] incorporated 10% APP into a whole grain bread formulation and detected markedly lower volume (1.864 ± 0.008 mL/g) of the enriched bread samples in comparison with the control (1.994 ± 0.008 mL/g), which is also in agreement with our study. Generally, it is known that dry APP hinders the expansion capacity of dough, most probably due to the interaction between gluten and APP fiber material or gluten dilution [31]. Many compounds present in different types of fruits are hydrophilic (with natural affinity for water), which reflects the increased viscosity of the bread formulations. Furthermore, decreased volume and increased firmness of bread loaves enriched with fruit ingredients (attributed to the increasing viscosity) may also be caused by the reduction of gluten amounts in the dough due to the partial supplementation of WF by these additions. Since the gluten network is mainly responsible for gas retention during fermentation, lower gluten content could lead to bread loaves with lower volume [9,32]. Another factor causing a reduction in the bread volume may be a negative effect of APP on gluten hydration and the formation of a gluten network due to competition for water between the fiber and the gluten protein [28]. Similar findings have also been reported in other studies [9,32] which investigated the addition of fruit powders to bread formulations.

Besides the quality attributes (including volume) of bread loaves, sensory characteristics are also important determinants of their acceptance by consumers. In such a manner, the acceptable amounts of additives for the product that consumers are willing to accept can be recommended [33]. High levels of phytochemicals in food products can significantly affect the taste and aroma of cereal-based products, in particular [34]. Although the addition of fruit by-products generally decreases the global acceptability of bakery products (such as bread) [10], no significant differences in various sensory parameters between our APP-containing bread loaves and those from the control group were observed. We assume that this phenomenon may be related to the addition of an adequate amount of the APP, which did not significantly affect the sensory quality of the analyzed bread loaves. To this regard, Alongi et al. [35] also did not find considerable changes in the sensory properties of biscuits in which WF was replaced by 8 to 14% APP. Based on the sensory and compositional attributes, Usman et al. [15] also found that cookies with good quality and improved organoleptic properties can be prepared through using 10% APP with WF. On the other hand, some changes in the sensory properties of cereal products were noted when a higher amount (15%) of APP was applied [36], which is in contrast with the results obtained by Vijayaragavi and Arivuchudar [37] showing high acceptability of bread with an addition of 20% AP. In summary, taking into account the results from our nutritional profile analysis along with the findings of sensory rating it can be concluded that 10% APP addition can be considered a valuable ingredient to be successfully used to develop novel formulations of wheat bread with health benefits.

## 4. Materials and Methods

### 4.1. Obtaining the Apple Pomace Powder

In the present research, different varieties of apple (Golden Delicious, Mutsu, Melodie, and Idared) were collected from the experimental plots of the Botanical Garden (Slovak University of Agriculture in Nitra, Nitra, Slovakia). Fresh AP as processing waste generated after apple juice manufacturing was used for dehydrated APP preparation. For this purpose, the AP sample was dried at 55 °C for 24 h using a laboratory hot-air dryer (Universal oven UF 160, Memmert GmbH + Co.KG, Büchenbach, Germany). Further, the dried AP was reduced to powder by a laboratory homogenizer (ETA Gratus 0028 90030, ETA-Slovakia Ltd., Bratislava, Slovakia), and uniform particle size (1 mm) was achieved by sieving. Subsequently, the powder was packaged in polyethylene bags and kept in a dark place at room temperature until testing.

### 4.2. Raw Materials

Wheat flour (WF; T-650) was obtained from a grinding mill (Mlyn Zrno Ltd., Veľké Hoste, Slovakia). Other raw materials, such as salt (Solné Mlýny JSC., Olomouc, Czech Republic), saccharose (Považský cukor JSC, Trenčianska Teplá, Slovakia), and compressed yeast (Dr. Oetker Ltd., Bratislava, Slovakia), were purchased at a local market.

### 4.3. Chemicals

All chemicals were of analytical grade, and were purchased from Sigma Aldrich (Saint Louis, MO, USA) and Reachem (Bratislava, Slovakia).

### 4.4. Flour Blend Preparation

Using different combinations of the available raw materials of the APP (0%, 1%, 2%, 5%, and 10%; *w*/*w*) and the WF, five flour combinations were prepared: control sample (0% APP) and four experimental samples (99% WF + 1% APP; 98% WF + 2% APP; 95% WF + 5% APP, and 90% WF + 10% APP).

### 4.5. Elaboration of Standard and Apple Pomace Powder-Enriched Bread Loaves

Bread samples (Figure 2) were prepared according to the methodology of Valková et al. [9], with some modifications. Five bread formulations were prepared using five flour blends, saccharose (1 g/100 g), salt (2 g/100 g), water (60 g/100 g), and yeast (2 g/100 g). In the first step, dry yeast was reactivated in a solution of saccharose at 32 °C for 5 min. After yeast reactivation, all the ingredients were mixed with a mixer (DIOSNA SP 12, DIOSNA Dierks & Söhne GmbH, Osnabrück, Germany) operating at two speeds (first speed: 4 min, 4.0 kW; second speed: 6 min, 8.0 kW) for 10 min until the gluten was developed. The dough was weighed and divided into 250 g pieces, which were rounded in a spherical shape by hand and put into an individual aluminium vessel. The fermentation step was carried out in a fermentation cabinet (MIWE cube, Pekass Ltd., Pilsen, Czech Republic) at 32 °C for 40 min. The loaves were baked in two phases (Laboratory oven MIWE cube, Pekass Ltd., Pilsen, Czech Republic). They were baked at 180 °C with the addition of 160 mL steam (at the same temperature) for 10 min (phase I) followed by baking at 210 °C for 7 min (phase II). The obtained bread loaves were cooled to ambient temperature for 2 h and prepared for analyses. In total, 15 bread loaves were produced for the examination (three bread loaves per experimental group).

### 4.6. Assessment of Samples Nutritional and Energetic Values

In this study, nutritional (proximate composition) analysis was performed in both the samples of raw materials (APP, WF) and experimental bread loaves. Within this group of measurements, the contents of moisture, ash, crude protein, fat, total carbohydrates, and energetic value were determined.

The moisture of all samples was detected using the moisture analyzer DBS 60 3 (Kern & Sohn GmbH, Altstadt, Germany). In this case, 1.0 g of each dried sample was weighed on the sample plate and measurement was performed at 120 °C for 10–15 min.

The ash was determined in accordance with AACC standard 08-01 [38]. In this context, the value for ash content was detected by incinerating the samples in a muffle furnace at 900 °C for 5 h.

In addition, the crude protein content was determined in accordance with AACC standard 08-01 [20]. Here, a nitrogen content was measured by the semi-micro Kjeldahl method after the three steps of digestion, distillation, and titration. The nitrogen value was corrected and multiplied by a factor of 6.25 to obtain the protein value.

The fat content (FC) was measured using the Ankom XT15 Fat Extractor (Ankom Technology, Fairport, NY, USA) in line with the instructions of the manufacturer with slight modification. About 1.5 g (W1) of the sample was weighted to the XT4 filter bag (Ankom, NY, USA), and then dried for 3 h in a laboratory hot-air oven (WTB, Binder, Germany) at 105 °C. Further, the samples were placed in a laboratory desiccator for 15 min and afterwards were re-weighed (W2) and extracted 60 min at 90 °C with petroleum ether. After extraction, the samples were dried again in a laboratory oven at 105 °C for 30 min, placed in a desiccator for 15 min, and re-weighed (W3). Finally, the FC (%) was calculated according to Formula (1):FC = [(W2 − W3)/W1] × 100(1)

Total carbohydrate content (TCC) was determined according to the following Formula (2) as reported by Arraibi et al. [1]:TCC (%) = 100 − moisture (%) − protein (%) − lipids (%) − ash (%)(2)

The corresponding energy was calculated according to the Atwater system (3) as reported by Arraibi et al. [1]:Energy (kcal/100 g) = 4 × (% proteins + % carbohydrates) + 9 × (% fat)(3)

### 4.7. Determination of Antioxidant Activity and Total Polyphenols Content

Before the analysis, ethanolic extracts were prepared from the raw materials (APP and WF) and all experimental bread samples. For each extraction, 0.2 g of APP and WF or 0.5 g of bread was extracted by 20 mL (APP, WF) or 40 mL (bread) of 80% ethanol for 2 h and centrifuged at 4000× *g* for 10 min in a Rotofix 32A (Hettich, Spenge, Germany). The supernatants were used for measurement of antioxidant activity (AA) using 2,2 diphenyl 1 picrylhydrazyl (DPPH method) and for detection of total polyphenols content (TPC).

Antioxidant activity of the raw materials (APP and WF) and experimental bread samples was measured according to the methodology described by Valková et al. [39]. Volumes of 0.4 mL of APP or WF extracts and 1 mL of bread sample extracts were added to 3.6 mL (APP, WF) or 4 mL (bread) of DPPH solution (0.025 g DPPH in 100 mL ethanol). Absorbance of the reaction mixture at 515 nm was determined using a Jenway 6405 UV/Vis spectrophotometer (Cole Parmer, Stone, UK). Antioxidant activity was expressed as Trolox equivalent antioxidant capacity (TEAC) in grams per kilogram of dry weight (dw).

Total polyphenols content was also established by the method of Valková et al. [39] using the Folin Ciocalteu (F-C) reagent. In this case, 0.1 mL of APP and WF extracts or 0.2 mL of bread extracts was mixed with 0.1 mL (APP, WF) or 0.2 mL (bread) of F-C reagent, 1 mL (APP, WF) or 2 mL (bread) of 200 g/L sodium carbonate and 8.8 mL of distilled water. After 30 min in darkness, the absorbance at 700 nm was measured using a Jenway 6405 UV/Vis spectrophotometer. Gallic acid was used as a standard, and the results were expressed as gallic acid equivalents (GAE) in grams per kilogram of dw.

### 4.8. Evaluation of Bread Loaves Volume

The volume of bread loaves was measured using a laser-based scanner (VolScanProfiler 300, Stable Micro Systems, Godalming, UK) as reported by Valková et al. [9].

### 4.9. Experimental Design of Bread Sensory Evaluation

A panel of 20 trained members of both sexes (10 male:10 female) was involved in the sensory rating of the analyzed bread samples. The rating questionnaire was prepared according to the study of Valková et al. [9]. In this concept, the analysis was carried out using semi-structured scales scoring 1 (lowest) to 5 (highest). The following sensory parameters were evaluated: shape and overall bread appearance, crust surface and its properties, the overall appearance of the crumb, aroma, taste, and overall acceptability of bread. For each of these parameters, the average response of the panellists was calculated by the following coefficients: shape and overall appearance—1, crust surface and properties—2, the overall appearance of the crumb—2, aroma—4, taste—5, and overall acceptability of bread sample—6. Finally, sensory properties were scored on the final scale from 1 to 100.

### 4.10. Statistical Analysis

The data was submitted to one-way analysis of variance (ANOVA) and the means were compared by the Tukey test at 5% of probability, using statistical software Prism 8.0.1 (GraphPad Software, San Diego, CA, USA). All analyses were performed in triplicate.

## 5. Conclusions

In the current study, the nutritional and antioxidant properties, as well as selected physico-chemical and sensory characteristics of bread samples supplemented with different concentrations of APP were investigated. From the results it is clearly evident that replacement of WF by APP in the wheat bread had a significant effect on its nutritional value, antioxidant potential, and volume. As compared to non-supplemented bread samples, the addition of APP to bread formulations resulted in increased ash, total carbohydrate content, and energetic values, and markedly lower protein and fat contents, and loaf volume, as well. On the other hand, sensory properties of the bread loaves were not demonstrably affected by APP supplementation in all concentrations used. The great benefit for the food industry mainly lies in the use of by-products from apple juice processing, reflecting in the reduction of the environmental burden, as well as saving economic resources in production. Our study revealed that APP can be recognized as a suitable ingredient for bakery purposes given its proven benefits, and the addition of 10% APP can be recommended for the production of bread with a rich nutritional profile and adequate quality characteristics.

## Figures and Tables

**Figure 1 plants-11-01256-f001:**
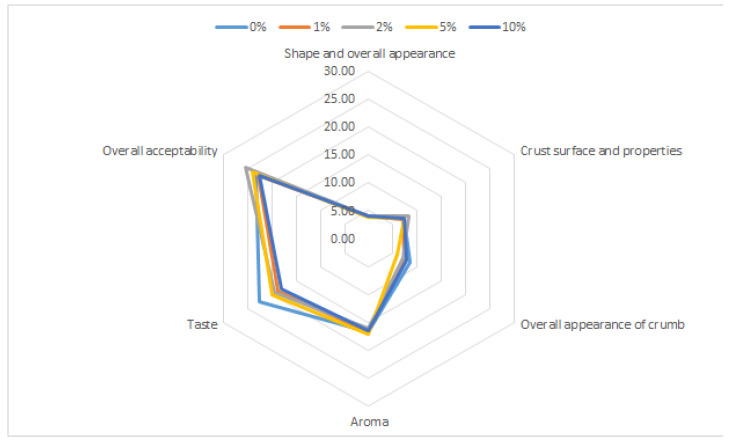
Radar plot obtained from sensory evaluation of the bread samples with different (0%, 1%, 2%, 5% and 10%) addition of APP.

**Figure 2 plants-11-01256-f002:**
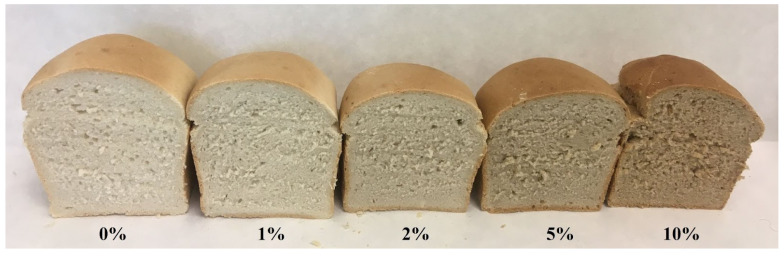
Prepared experimental groups of bread loaves. 0%: control sample, 1%: 1% apple pomace powder addition, 2%: 2% apple pomace powder addition, 5%: 5% apple pomace powder addition, 10%: 10% apple pomace powder addition.

**Table 1 plants-11-01256-t001:** Nutritional analysis of the apple pomace powder and wheat flour.

Parameters	APP	WF
Moisture (%)	6.55 ± 0.16 ^a^	12.13 ± 0.19 ^b^
Ash (%)	1.65 ± 0.08 ^a^	0.61 ± 0.05 ^b^
Protein (%)	2.18 ± 0.09 ^a^	11.78 ± 0.13 ^b^
Fat (%)	1.58 ± 0.01 ^a^	2.01 ± 0.06 ^b^
Carbohydrate (%)	88.04 ± 0.09 ^a^	73.47 ± 0.11 ^b^
Energetic value (kcal/100 g)	375.10 ± 0.07 ^a^	359.09 ± 0.09 ^b^
TPC (g/kg)	9.53 ± 0.03 ^a^	0.41 ± 0.02 ^b^
AA (g/kg)	7.70 ± 0.17 ^a^	0.16 ± 0.03 ^b^

Mean ± standard deviation. Values with different superscripts within the same row are significantly different (*p* < 0.05). APP—apple pomace powder, WF—wheat flour, TPC—total polyphenols content expressed as grams of gallic acid equivalents per kilogram dry weight, AA—antioxidant activity expressed as grams of Trolox equivalents per kilogram dry weight.

**Table 2 plants-11-01256-t002:** Nutritional analysis of the experimental bread loaves.

Parameters	Additions of APP (%)				
	0	1	2	5	10
Moisture (%)	13.69 ± 0.36 ^a^	13.74 ± 0.30 ^a^	13.72 ± 0.33 ^a^	13.67 ± 0.23 ^a^	13.15 ± 0.29 ^a^
Ash (%)	0.54 ± 0.03 ^a^	0.65 ± 0.02 ^b^	0.70 ± 0.03 ^b^	0.83 ± 0.04 ^c^	0.95 ± 0.02 ^d^
Protein (%)	17.40 ± 0.07 ^a^	16.18 ± 0.08 ^b^	16.08 ± 0.05 ^b^	13.15 ± 0.07 ^c^	12.73 ± 0.09 ^d^
Fat (%)	6.72 ± 0.06 ^a^	6.60 ± 0.03 ^b^	6.54 ± 0.05 ^b^	6.00 ± 0.07 ^c^	5.82 ± 0.02 ^d^
Carbohydrate (%)	61.65 ± 0.11 ^a^	62.83 ± 0.08 ^b^	62.96 ± 0.11 ^b^	66.35 ± 0.09 ^c^	67.35 ± 0.21 ^d^
Energetic value (kcal/100 g)	376.68 ± 0.10 ^a^	375.44 ± 0.08 ^b^	375.02 ± 0.15 ^c^	372.00 ± 0.12 ^d^	372.70 ± 0.19 ^e^
TPC (g/kg)	0.60 ± 0.07 ^a^	0.99 ± 0.02 ^b^	1.12 ± 0.04 ^c^	2.18 ± 0.02 ^d^	2.56 ± 0.03 ^e^
AA (g/kg)	1.65 ± 0.08 ^a^	1.99 ± 0.07 ^b^	2.14 ± 0.03 ^c^	2.61 ± 0.05 ^d^	2.79 ± 0.02 ^e^

Mean ± standard deviation. Values with different superscripts within the same row are significantly different (*p* < 0.05). 0%: control sample, 1%: 1% apple pomace powder addition, 2%: 2% apple pomace powder addition, 5%: 5% apple pomace powder addition, 10%: 10% apple pomace powder addition, TPC—total polyphenols content expressed as grams of gallic acid equivalents per kilogram dry weight, AA—antioxidant activity expressed as grams of Trolox equivalents per kilogram dry weight.

**Table 3 plants-11-01256-t003:** Volume of the experimental bread samples.

Additions of APP (%)	Volume (mL)
0	610.42 ± 2.05 ^a^
1	542.77 ± 1.88 ^b^
2	510.07 ± 1.09 ^c^
5	495.12 ± 1.55 ^d^
10	482.77 ± 2.64 ^e^

Mean ± standard deviation. Values with different superscripts within the same column are significantly different (*p* < 0.05). 0%: control sample, 1%: 1% apple pomace addition, 2%: 2% apple pomace addition, 5%: 5% apple pomace addition, 10%: 10% apple pomace addition.

## Data Availability

Data available in a publicly accessible repository.

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
