# Peer review of "Selected Physico-Chemical, Nutritional, Antioxidant and Sensory Properties of Wheat Bread Supplemented with Apple Pomace Powder as a By-Product from Juice Production"

_plants, 2022, doi:10.3390/plants11091256_

Round 1

Reviewer 1 Report

The study was very well presented and enhances our knowledge about the use of apple pomace powder in bread formulations.

Author Response

Reviewer #1

The study was very well presented and enhances our knowledge about the use of apple pomace powder in bread formulations.

Response: Thank you very much for the favorable comment. We really appreciate it.

Reviewer 2 Report

In my opinion, the article is not exceptionally innovative because research on the addition of apple pomace to bread making has already been done. However, as the authors noted, this additive caused unfavorable changes in the characteristics of bread, so it is undoubtedly worthwhile to conduct research in the direction of optimization of bakery recipes. In the aim of the research, it is necessary to emphasize more what is new in this article.

Apple pomace is a waste product and its use in baking is justified. The literature review and discussion should be carried out even more thoroughly because several important publications (see below) on similar topics were omitted.

Table 2. Please check the content of carbohydrates in bread with 1%, there is higher value than in bread with 2% - please explain why, because it seems to be a mistake because there were more carbohydrates in apple pomace than in wheat flour. It is generally seen that the value increases with higher amounts of this additive. The section Flour Blend Preparation should be placed in the baking section so it will be easier to read.I do not understand the adopted order of chapters, in my opinion the chapter Elaboration of Standard and Apple Pomace Powder-Enriched Bread Loaves should be in place of the section flour blend preparation. First, the bread was baked and then it was evaluated.In addition to the mixing time of 10 min. Please provide more details about the mixer, at what speed was it mixed?

The section titled Data Processing should be called Statistical Analysis 

Proposed references

Lu, Q., Liu, H., Wang, Q., & Liu, J. (2017). Sensory and physical quality characteristics of bread fortified with apple pomace using fuzzy mathematical model. International Journal of Food Science and Technology52(5), 1092–1100. https://doi.org/10.1111/ijfs.13280

He, Y., & Lu, Q. (2015). Impact of apple pomace on the property of french bread. Advance Journal of Food Science and Technology8(3), 167–172. https://doi.org/10.19026/ajfst.8.1487

Djeghim, F., Bourekoua, H., Różyło, R., Bieńczak, A., Tanaś, W., & Zidoune, M. N. (2021). Effect of by-products from selected fruits and vegetables on gluten-free dough rheology and bread properties. Applied Sciences (Switzerland)11(10). https://doi.org/10.3390/app11104605

Jannati, N., Hojjatoleslamy, M., Hosseini, E., Mozafari, H. R., & Siavoshi, M. (2018). Effect of apple pomace powder on rheological properties of dough and Sangak bread texture. Carpathian Journal of Food Science and Technology10(2), 77–84.

Masoodi, F. A., & Chauhan, G. S. (1998). Use of apple pomace as a source of dietary fiber in wheat bread. Journal of Food Processing and Preservation22(4), 255–263. https://doi.org/10.1111/j.1745-4549.1998.tb00349.x

Bchir, B., Rabetafika, H. N., Paquot, M., & Blecker, C. (2014). Effect of Pear, Apple and Date Fibres from Cooked Fruit By-products on Dough Performance and Bread Quality. Food and Bioprocess Technology7(4), 1114–1127. https://doi.org/10.1007/s11947-013-1148-y

Author Response

Reviewer #2

Point 1: In my opinion, the article is not exceptionally innovative because research on the addition of apple pomace to bread making has already been done. However, as the authors noted, this additive caused unfavorable changes in the characteristics of bread, so it is undoubtedly worthwhile to conduct research in the direction of optimization of bakery recipes. In the aim of the research, it is necessary to emphasize more what is new in this article.

Response: We clarified and modified the goal of the research directly in the manuscript.

Point 2: Apple pomace is a waste product and its use in baking is justified. The literature review and discussion should be carried out even more thoroughly because several important publications (see below) on similar topics were omitted.

Proposed references

Lu, Q., Liu, H., Wang, Q., & Liu, J. (2017). Sensory and physical quality characteristics of bread fortified with apple pomace using fuzzy mathematical model. International Journal of Food Science and Technology52(5), 1092–1100. https://doi.org/10.1111/ijfs.13280

He, Y., & Lu, Q. (2015). Impact of apple pomace on the property of french bread. Advance Journal of Food Science and Technology8(3), 167–172. https://doi.org/10.19026/ajfst.8.1487

Djeghim, F., Bourekoua, H., Różyło, R., Bieńczak, A., Tanaś, W., & Zidoune, M. N. (2021). Effect of by-products from selected fruits and vegetables on gluten-free dough rheology and bread properties. Applied Sciences (Switzerland)11(10). https://doi.org/10.3390/app11104605

Jannati, N., Hojjatoleslamy, M., Hosseini, E., Mozafari, H. R., & Siavoshi, M. (2018). Effect of apple pomace powder on rheological properties of dough and Sangak bread texture. Carpathian Journal of Food Science and Technology10(2), 77–84.

Masoodi, F. A., & Chauhan, G. S. (1998). Use of apple pomace as a source of dietary fiber in wheat bread. Journal of Food Processing and Preservation22(4), 255–263. https://doi.org/10.1111/j.1745-4549.1998.tb00349.x

Bchir, B., Rabetafika, H. N., Paquot, M., & Blecker, C. (2014). Effect of Pear, Apple and Date Fibres from Cooked Fruit By-products on Dough Performance and Bread Quality. Food and Bioprocess Technology7(4), 1114–1127. https://doi.org/10.1007/s11947-013-1148-y

Response: The sections were modified and supplemented with new information directly in the manuscript.

Point 3: Table 2. Please check the content of carbohydrates in bread with 1%, there is higher value than in bread with 2% - please explain why, because it seems to be a mistake because there were more carbohydrates in apple pomace than in wheat flour. It is generally seen that the value increases with higher amounts of this additive.

Response: Thank you for your attention regarding polysaccharide values. Indeed, the mistake occurred in the data. We thoroughly checked the measured values and corrected any erroneous values directly in the manuscript.

Point 4: The section Flour Blend Preparation should be placed in the baking section so it will be easier to read. I do not understand the adopted order of chapters, in my opinion the chapter Elaboration of Standard and Apple Pomace Powder-Enriched Bread Loaves should be in place of the section flour blend preparation. First, the bread was baked and then it was evaluated. In addition to the mixing time of 10 min. Please provide more details about the mixer, at what speed was it mixed?

Response: The order of the chapters has been changed and the information has been added directly in the manuscript.

Point 5: The section titled Data Processing should be called Statistical Analysis. 

Response: Edited directly in the manuscript.

Reviewer 3 Report

I will reject this manuscript because of following reasons:

1.In the reference 27, the addition of 8 to 14% APP did not find considerable changes in sensory properties, >15% can changed the sensory properties. However, 10% APP addition was valuable---, how author confirmed that 10% was the best formula.

2.There are lots of paper discussed the addition of by-product into bakery products, especially for apple pomace or powder. What is the main point? just sensory? antioxidant?

3.Why the author used "selected" to describe this manuscript? And there are lots of results have problems, for example: the standard of antioxidant test? the IC50 of antioxidant? the extraction protocol for extracting sample into antioxidant test?

Author Response

Reviewer #3

I will reject this manuscript because of following reasons:

Point 1: In the reference 27, the addition of 8 to 14% APP did not find considerable changes in sensory properties, >15% can changed the sensory properties. However, 10% APP addition was valuable---, how author confirmed that 10% was the best formula.

Response: Directly revised in the manuscript.

Point 2: There are lots of paper discussed the addition of by-product into bakery products, especially for apple pomace or powder. What is the main point? just sensory? antioxidant?

Response: The main goal of our study was to effectively utilize the by-product of apple juice production (i.e., apple pomace) realized in our research centre in large scale from apples growing on our experimental field for further practical application to eliminate the amounts of the waste product in the environment. To determine the most adequate concentration of apple pomace powder used for bread supplementation meeting health-promoting nutrition with the preservation of acceptable physical and sensory properties of bread loaves, the standard methods for such evaluations were employed. 

Point 3: Why the author used "selected" to describe this manuscript? And there are lots of results have problems, for example: the standard of antioxidant test? the IC50 of antioxidant? the extraction protocol for extracting sample into antioxidant test?

Response: The term “selected” was used because of selected standard analyzes employed in the study. Based on them, the most recommended concentration of apple pomace powder (10%) will be used in our next experimental research activities to gain more comprehensive results from this issue. For instance, the bread supplemented with 10% of apple pomace powder will be further evaluated from chemical composition, macro- and micro-elements, textural properties, etc. points of view.

As a standard of antioxidant test, Trolox was used.  We did not determine the IC50 of antioxidant activity in our study. Instead, we used a methodological procedure to determine the values for antioxidant activity in units of grams of Trolox equivalents per kilogram dry weight. We have considered the determination of the antioxidant activity in such way on the basis of several studies focused on this issue, in which the IC50 was also not expressed (Rana et al., 2015; Ivanišová et al., 2017; Grygorieva et al., 2020). Anyway, thanks for the advice, we will also consider determining the IC50 for more complex results in the future.

The extraction procedure is given directly in the manuscript and is adapted to this type of samples (plant material and cereal products). In addition, we have verified this extraction method in several experimental works that have already been published on similar issues (Valková et al., 2021a; Valková et al. 2021b; Valková et al., 2021c).

References:

Rana, S., Gupta, S., Rana, A., & Bhushan, S. (2015). Functional properties, phenolic constituents and antioxidant potential of industrial apple pomace for utilization as active food ingredient. Food Science and Human Wellness, 4(4), 180-187.

Ivanišová, E., Grygorieva, O., Abrahamova, V., Schubertova, Z., Terentjeva, M., & Brindza, J. (2017). Characterization of morphological parameters and biological activity of jujube fruit (Ziziphus jujuba Mill.). Journal of Berry Research, 7(4), 249-260.

Grygorieva, O., Klymenko, S., Vergun, O., Mňahončáková, E., Brindza, J., Terentjeva, M., & Ivanišová, E. (2020). Evaluation of the antioxidant activity and phenolic content of Chinese quince (Pseudocydonia sinensisSchneid.) fruit. Acta Scientiarum Polonorum Technologia Alimentaria, 19(1), 25-36.

Valková, V., Ďúranová, H., Ivanišová, E., Kravárová, A., Hillová, D., & Gabríny, L. (2021a). Influence of variety on total polyphenols content and antioxidant activity in apple fruits (Malus domestica Borkh.). Agrobiodiversity for Improving Nutrition, Health and Life Quality, 5(2).

Valkova, V., Ďúranová, H., Miškeje, M., Ivanišová, E., Gabriny, L., & Kačániová, M. (2021b). Physico-chemical, antioxidant and microbiological characteristics of bread supplemented with 1% grape seed micropowder. Journal of Food & Nutrition Research, 60(1).

Ďúranová, H., Ivanišová, E., Galovičová, L., Godočíková, L., Borotová, P., Kunová, S., ... & Mňahončáková, E. (2021c). Antioxidant and antimicrobial activities of fruit extracts from different fresh chili peppers. Acta Scientiarum Polonorum Technologia Alimentaria, 20(4), 465-472.

Round 2

Reviewer 3 Report

The author had corrected the manuscript according to my comments.